# Advanced Clinical Practitioners in Primary Care in the UK: A Qualitative Study of Workforce Transformation

**DOI:** 10.3390/ijerph17124500

**Published:** 2020-06-23

**Authors:** Catrin Evans, Ruth Pearce, Sarah Greaves, Holly Blake

**Affiliations:** 1School of Health Sciences, University of Nottingham, Nottingham NG7 2HA, UK; ruth.pearce@nottingham.ac.uk (R.P.); sarah.greaves@nottingham.ac.uk (S.G.); holly.blake@nottingham.ac.uk (H.B.); 2NIHR Nottingham Biomedical Research Centre, Nottingham NG7 2HA, UK

**Keywords:** workforce, advanced clinical practice, primary care, workplace wellness, general practice

## Abstract

Escalating costs and changing population demographics are putting pressure on primary care systems to meet ever more complex healthcare needs. Non-medical ‘advanced clinical practitioner’ (ACP) roles are increasingly being introduced to support service transformation. This paper reports the findings of a qualitative evaluation of nursing ACP roles across General Practices in one region of the UK. Data collection involved telephone interviews with 26 participants from 3 different stakeholder groups based in 9 practice sites: ACPs (*n* = 9), general practitioners (*n* = 8) and practice managers (*n* = 9). The data was analysed thematically. The study found a high degree of acceptance of the ACP role and affirmation of the important contribution of ACPs to patient care. However, significant variations in ACP education, skills and experience led to a bespoke approach to their deployment, impeding system-wide innovation and creating challenges for recruitment and ongoing professional development. In addition, a context of high workforce pressures and high service demand were causing stress and there was a need for greater mentorship and workplace support. System wide changes to ACP education and support are required to enable ACPs to realise their full potential in primary care in the UK.

## 1. Introduction

Across the world, health systems are increasingly challenged by the complex needs of ageing populations, health workforce shortages and escalating costs [1,2]. Amidst these challenges, there is a pressing need to develop effective and efficient primary health services to contribute to the global goal of universal health coverage [3]. In many countries, including the UK, current debates focus on achieving the optimum skill mix within primary care and the optimum models of service provision [4,5].

In the UK, health has been a devolved responsibility of its four countries (England, Scotland, Wales and Northern Ireland) since the late 1990s and each country has its own ‘National Health Service’ (NHS). The NHS in each country are tax-funded services, and, although some policy differences exist, they share many similar goals and challenges [6].

Primary care in the UK has traditionally been heavily dependent upon General Practitioners (GPs), medical doctors who work on their own or in groups/hubs referred to as ‘Practices’ or ‘General Practices’. GPs operate as businesses and can either be partners in a practice or salaried employees. Primary care services are commissioned from a practice by geographically based ‘Clinical Commissioning Groups’ (CCGs) funded by the NHS [7]. The structure and organisation of practices can be variable. At the minimum, most employ nursing staff (e.g., Practice Nurses, Nurse Practitioners or Advanced Nurse Practitioners) and ancillary staff (e.g., Phlebotomists or Healthcare Assistants), as well as Receptionists and a Practice Manager (PM). GP practices link with a wide range of other primary care services (e.g., dental, pharmacy, physiotherapy, podiatry, audiology) to provide comprehensive services to their local populations. In some settings (especially cities), in addition to services delivered through GP practices, primary care is also available through a system of ‘Walk-in’ Centres, Family Planning/Sexual Health Clinics or ‘Out of Hours’ Urgent Care Centres. Regardless of the service mode, for patients, primary care is free at the point of delivery.

Several recent reports have highlighted a severe and growing shortage of GPs in the UK, and a need to re-think traditional models of primary care service delivery in response to demographic changes and evolving healthcare needs [8,9]. The NHS Long Term Plan [10] and General Practice Forward View [11] have proposed new, more integrated models of care that span traditional boundaries between primary and secondary care and between health and social care [12,13,14]. These proposals are supported by an interim NHS People Plan [15] that articulates an urgent need to invest in the development of new non-medical clinical roles and, in particular, advanced level skills to enable workforce expansion [15,16,17,18]. Working with Health Education England (HEE), a national non-departmental public body that supports NHS workforce development), the interim NHS People Plan aims to define sets of advanced skills to apply across a wide range of professional groups by developing advanced clinical practice (ACP) roles with primary and community services targeted for greater service and workforce expansion [15]. ACP roles are being developed across a range of professions in multiple sectors [19]. However, in primary care, by far the largest non-medical cadre is nursing [9].

The development of ACP roles in nursing has varied globally and is referred to by varying nomenclature (e.g., in some constituencies, roles are referred to as ‘nurse practitioner’ and in others as ‘advanced nurse practitioner’) [20]. The United States and Canada established the role of the nurse practitioner in the mid-1960s to provide primary care to populations in rural and remote areas. In North America, advanced practice roles are well established. They require a Master’s degree, have a protected title and are licensed and regulated separately [20,21,22]. In the UK, by contrast, advanced nursing roles have developed since the 1980s but in a more organic manner [20,23]. Rather than being guided by national policy, developments have been locally driven and reactive, responding to the particular needs of diverse employers or regional healthcare entities with a lack of consistency and lack of standardization in role specification, competencies, educational background or role title [20,23]. For example, one UK study identified 595 different job titles being used for specialist and advanced practice [24]. Titles refer to widely differing roles which are underpinned by differing sets of qualifications and competencies [23]. To date, there has been no standardized educational route into advanced practice roles [25]. This situation has led to a lack of clarity for employers, patients and fellow healthcare workers [23,24,26].

In order to bring stability to workforce developments in this area, in 2017, HEE published a ‘Multiprofessional Framework for Advanced Clinical Practice’ for England [27] that sought to provide a clear definition of ACP. The framework stated that: “*Advanced clinical practice is delivered by experienced, registered health and care practitioners. It is a level of practice characterised by a high degree of autonomy and complex decision-making. This is underpinned by a masters level award or equivalent that encompasses the four pillars of clinical practice, leadership and management, education and research, with demonstration of core capabilities and area specific clinical competence.”* [27] In this definition, ACP is understood as a level of practice rather than a specific role. A key distinguishing feature of ACP is the level of autonomy exercised by a practitioner as well as an ability to operate at this autonomous advanced level across 4 domains, including, but not limited to, clinical practice. These are referred to as the 4 ‘pillars’ of ACP (education, leadership, research and clinical practice), and the framework describes a set of generic core capabilities that should be achieved within each pillar. The HEE ACP framework represents an important step forward by providing an overarching structure to align existing practice and education by creating greater consistency across ACP workforce developments.

The HEE ACP framework applies specifically to England, but it has been developed in consultation with stakeholders that represent professions across the UK (e.g., Professional Bodies and Royal Colleges) and has drawn upon similar advanced practice frameworks that exist in the other 3 countries [28,29,30]. The release of the framework aims to support NHS providers to enable the delivery of sustainable health and care services. It also recognizes that introducing, developing and supporting ACP within an organisation requires good governance in order to embed ACP into the workplace [27]. National work is also ongoing to establish consistent processes and standards for advanced practice education and a credentialing system.

In some specialties, additional work has taken place to define specialty- or sector-specific capabilities within the 4 pillars. Within primary care, a framework for nurses working as ACPs was published in 2020 [31].

As work proceeds in the UK to promote the role of ACPs within the primary care workforce transformation agenda, there is a need to evaluate these roles. With respect to advanced nursing roles specifically, systematic reviews of national and international evidence demonstrate significant benefits to patient care and system-wide outcomes within primary care [32,33,34,35,36,37,38]. In terms of the ‘clinical’ pillar of ACP, across varied populations and settings, the evidence consistently shows that advanced nurses are able to provide safe and effective clinical care and achieve high levels of patient satisfaction. Less well understood, however, are the impacts and outcomes associated with the other 3 pillars of ACP in primary care (education, research and leadership). Likewise, there is a relative paucity of evidence regarding the factors that influence the acceptance and integration of ACPs into primary care organisations, the impact they have on wider organisational outcomes and workloads or the educational and support needs of ACPs to promote job satisfaction and retention [39]. In order to promote the wide scale roll-out of ACPs in primary care, these issues require greater investigation.

### Aims and Objectives

This paper reports on a project that aimed to explore how ACP nursing roles were being implemented within GP practices in primary care in a single region of the UK. The specific objectives were: (i) to explore ACP role implementation from a range of stakeholder perspectives (ACPs, Practice Managers and GPs); (ii) to identify key barriers and facilitators to ACP role implementation and sustainability; and (iii) to describe the perceived outcomes and impacts of ACP towards service transformation in primary care.

## 2. Materials and Methods

The project was commissioned by HEE (East Midlands region). It was designed as a formative evaluation [40,41,42] in order to inform policy regarding the roll-out and potential scale-up of ACPs within primary care GP practices [43,44]. A qualitative descriptive approach [45,46] was adopted to explore the views and experiences of three staff groups in order to develop a holistic understanding of ACP role implementation: (i) ACP nurses, (ii) practice managers and (iii) GPs.

The evaluation took place in two counties (referred to as county A and County B) within the East Midlands region of the UK [47]. The counties were selected as sites for the evaluation because they represented a mix of urban, semi-urban and rural locations and had GP practices that were known to be employing ACPs.

Sampling was practice-based, utilising a purposive approach in order to identify practices of different sizes and locations and who employed ACPs. The identification of potentially suitable practices was facilitated by an ACP key informant who had a strategic role in ACP development at HEE. Practice managers (PMs) were telephoned by the project researcher (SG) and permission was obtained from each participating practice prior to approaching individual staff members. Once the relevant individuals had been identified, the project researcher contacted them by email or phone to explain the study. For those individuals who consented to take part, a mutually convenient date and time was agreed. Data were collected between March and June 2019 using in-depth semi-structured interviews [48]. All participants were provided with an information sheet (Appendix A) and were asked to sign and return a consent form (Appendix A). The interviews were digitally recorded and transcribed. All data were securely stored in a password-protected database. The interviews all followed a question guide (see Appendix A) but also allowed for free discussion of other topics by the respondents.

The interview transcripts were imported into a qualitative analysis software (NVIVO version 12 Pro [49]) and analysed using thematic analysis [50]. This followed several steps: (i) familiarisation through multiple readings of the transcripts, (ii) coding (assigning labels/codes to meaning units within the data), (iii) combining the codes into sub-themes and then (iv) interpreting and defining over-arching themes. The coding schema was influenced by the original evaluation questions (as reflected in the interview schedule), but also incorporated an inductive process in which new concepts were identified, interpreted and analysed to form sub-themes or themes. Particular attention was paid to similarities and differences within and between stakeholder groups and within and between practices [51].

The rigor of data analysis was enhanced by using a team approach [52]. Four members of the team (SG, CE, HB and RP) each read all the transcripts and debated, discussed and agreed to the final coding framework and key themes. The all-female project team had significant prior experience in health research and ACP. SG is a researcher and higher education administrator with experience in primary care settings, CE is a nurse educator with a research interest in health workforce capacity development, RP is a nurse educator and internationally recognized expert in ACP across all sectors and HB is a behavioural psychologist and health services researcher.

### Ethical Issues

The project was discussed with the institutional research governance team who classified it as a service evaluation according to the UK’s Health Research Authority Guidance [53]. In the UK, projects deemed to be evaluations of NHS services do not require approval from an ethical review board. Nonetheless, all principles of research integrity and good ethical conduct were followed. All participants received an information sheet explaining the study and providing details of who to contact in case of any concerns (Appendix A). It was stressed that participation in the project was completely voluntary, that interview content would be completely confidential and all data would be anonymized prior to reporting. All those who agreed to be interviewed returned a consent form (either by email or by post) prior to the telephone interview (Appendix A).

## 3. Results

### 3.1. Sample

Nine GP practices agreed to participate in the evaluation. In all except one practice, 3 individuals were interviewed (1 GP, 1 ACP and 1 PM). Hence, there were a total of 26 individuals in the sample: 8 GPs (4 male, 4 female), 9 ACPs (7 female, 2 male) and 9 PMs (7 female, 2 male). One GP was unavailable to participate due to sickness despite having previously agreed, and their colleagues declined to participate, citing excessive workload as reasons (the recruitment of GPs is a commonly cited challenge in the literature on primary care [54]).

The size of the practice (estimated by the number of patients on their list) varied from 5800 to 40,000. The number of GPs per practice ranged from 1 (with 2 locums) to 20. Practice governance was also varied. It was undertaken by practice partners for 6 of the practices, and by third party ‘not for profit’ providers for 3 of the practices. Two of the practices included ACPs as practice partners.

The majority of the practices had 1 or 2 ACPs (according to size of practice). One of the practices had 1 ACP doing ad-hoc sessions only. Some larger practices had additional ACPs in-training (up to 4), and the largest practice had 3 ACPs.

Staffing mix varied according to size of practice. Other healthcare staff included healthcare assistants (1–3 per practice), phlebotomists (1 per practice), community matron (1 per practice), clinical pharmacists (1 per practice) and practice nurses (1–18 per practice).

Practice settings reflected the diversity found in most UK regions [11,55,56], and included rural, semi-rural and urban settings (including suburbs and city centres). Catchment areas included areas of affluence, as well as areas of significant deprivation. Patient populations were highly diverse in terms of socio-economic and ethnic background. Some practices highlighted significant issues among their catchment demographic with chronic conditions (including COPD, asthma and diabetes), drug and alcohol misuse, mental health, domestic violence and safeguarding issues. See Table 1 for an overview of the 9 GP practices.

### 3.2. Advanced Clinical Practitioner Roles within the GP Practices

The ACPs were all nurses and had diverse prior experiences in primary and secondary care settings. Three ACPs reported managerial responsibilities (two were practice partners and one was a manager of a team of nurses, including other ACPs). The duration of their ACP experience varied from 6 months to 16 years—some had been recently appointed in their current practice (within one year) but had held ACP posts elsewhere previously, while some reported that the role was their first post in primary care.

ACPs undertook their own clinics; some conducted home visits and visits to care homes. Workloads per ACP varied greatly with numbers of patients seen per day by ACPs ranging from 22 to 57. There were some variations in the length of ACP appointment times according to the number of patents seen per day, and for some ACPs, this included telephone appointments. On average, the length of ACP appointments was 15 min (×4 practices), 10 min (×4 practices) and 12.5 min (×1 practice), or 5 min for telephone appointments (×1 practice).

### 3.3. Themes

Five overarching and interlinked themes were identified from this data. Figure 1 depicts these visually.

#### 3.3.1. Rationale for the ACP Role: Divergent Agendas

This theme explores the rationale provided for the ACP role from the perspective of the individual ACPs and their colleagues, revealing diverging agendas. GPs and PMs overwhelmingly reported the significance of the ACP role in terms of ensuring the sustainability of general practice in the face of significant monetary pressures and GP recruitment challenges:


**14PM:**
*We need to be making this move away from relying on GPs so much because they are not around, you know—they are not there to employ.*


The ACP role was widely seen to be a cost-effective role, critical for practice sustainability:


**23PM:**
*We have always pushed back a bit from going down that route until I guess we almost felt forced into it because of the situation with GP recruitment. We decided that we could probably look to have two ACPs for the price of a GP. But it has been a very positive experience.*


Several GPs and PMs suggested that they had lacked familiarity with the ACP role. However, as indicated in the quote above, once they had the experience of their skill set, the potential of ACPs to support service transformation became a more salient factor in recruitment decisions, noting that ACPs offered practices more flexibility in terms how services were offered and by whom:


**3GP:**
*A fully trained and skilled ACP can essentially do in general practice most, if not almost all, of everything that a GP can do.*


Some practices, however, noted that their ability to employ ACPs and to utilize them in innovative ways was hindered by their (small) size. Local plans to establish new ‘primary care networks’ were seen to offer more potential to support service development:

**5GP**: *At the moment a lot of general practices are just too small; it is not necessarily that we don’t need it [ACP], yes we definitely would love to have them do our home visits especially at care homes but we would only be sending them out three times a day, which is a completely waste. As a primary care network, we could get together and say right, let’s all get together and employ these ACPs and we could share that expertise.*

In contrast to the pragmatic rationale for embracing the ACP role given by GPs and PMs, for ACPs, the role was seen as an opportunity for personal career enhancement and progression, enabling practitioners to remain patient facing, develop an advanced skillset and a higher level of autonomous clinical practice, in contrast to more restrictive pathways of progression into managerial roles:


**22ACP:**
*It gives us as nurses the opportunity to flourish, learn more, keep going, without actually just being siphoned into a management role. In general practice for many, many years, people would come in and be a practice nurse and it would just stop, and I feel as though this is almost punching through that glass ceiling.*


ACPs rejected the notion of acting as a GP substitute, stressing the value that their nursing background brought to the role:

**1ACP**: *I don’t like the thought that nurse clinicians or ACPs are a cheap option of a GP because I believe that we bring a completely different way of working to our role and although we may be seeing similar patients, I think how we diagnose, how we treat, how we listen is very different to how GPs are taught.*

#### 3.3.2. Enactment of the ACP Role: The Four Pillars

This theme explores the extent to which the participants viewed the ACP role, or utilized the role, in line with the 4 pillars of advanced practice [27]. ACPs reported their work to be primarily clinically focused, with relatively less scope for development in other domains of advanced practice. This was attributed to the high demands of patient appointments, which varied considerably from reports of 22–57 appointments per day. In the main, ACPs tended to be allocated acute conditions and urgent ‘on the day’ appointments to help manage these high demands on general practice, leaving complex cases to General Practitioners (GPs):


**2PM:**
*We’ve tried to sort of focus their sessions on more urgent appointments—so on the day stuff, so that is definitely helping in terms of managing that side of things.*


However, in some practices, the ACPs’ caseload was similar to that of the GPs at the practice—this applied particularly to ACP partners and long-established ACPs, and several ACPs undertook the responsibility for care home visits, telephone triage and home visits. However, there were significant differences reported in the scope of clinical practice within the ACP role, with ACP deployment dependent upon their level of experience, competencies and skill set.

The ability of ACPs to operate autonomously was also shaped by national legal restrictions. For example, ACPs currently cannot sign death certificates or medical ‘fit-notes’. All stakeholders reported significant frustrations that GP authorizations were required for such tasks which created bottlenecks and inefficiencies in the service. Similarly, a major frustration for many respondents related to locally specific restrictions which varied according to local protocols, for example not being able to undertake telephone triage or being unable to request certain diagnostic tests or imaging:


**8ACP:**
*I can’t request x-rays or MRI, or even ultrasound which is frustrating so at the moment if I want to, I have to get the medical secretary to create the request on my behalf and get one of my GP colleagues to authorize it, so that is a frustration—it interrupts the flow of consultation and can add delays.*


In terms of ACPs’ leadership and management roles, here too, there was variability across the practices. More than half the ACPs reported having a significant leadership and management remit within their role. Two of these were ACP partners within their practices, and, as such, presented relatively unique partnership models and both had responsibility for ACP recruitment and annual appraisal. Two ACPs had direct line management responsibility for the practice nursing teams, including less experienced ACPs and healthcare assistants. Beyond the practice level, two ACPs were playing a regional role in ACP strategy and training development:


**8ACP:**
*I have been having a strategic lead role in advanced practice for the last couple of years so I established an ACP strategy group as part of the work force governance structure and that has led to some focused pieces of work around education supervision, competency framework; and that has been, you know, fairly slow and at times difficult journey but I think we have moved things along quite significantly. I also provide support regarding looking at service transformation as we’re moving into the primary care networks and looking at some of the more specialist services that we offer.*


Research capabilities and research-related activity appeared to be limited in the ACP group, due to restrictions on time and limited opportunities, with just a few ACPs reporting a contribution to research audits. The limited involvement in research appeared to be partly due to a perception that the ACP role was primarily for clinical work:


**25ACP:**
*At the moment, from the practice perspective they want me to see patients and that is essentially what I want to do and what I like doing so I guess it works.*


In terms of the ‘education’ pillar’, most of the ACPs were involved in educational support or mentorship of other staff. In some practices, this took place in an informal capacity, whereas in other practices, their educational role was more formally recognized:


**18GP:**
*This happens in the background, and so yes to be honest, the practice nurses, treatment room nurses, long term condition nurses, would all be standing outside the door of one of them to ask them—so yes they will ask for advice from ACP colleagues, they look up to them for advice and support.*


#### 3.3.3. Training and Support for the ACP Role: A Bespoke Picture

This theme explores the educational preparation and support for ACPs, revealing that their backgrounds and training experiences were highly individualised and variable, necessitating complex and bespoke recruitment, deployment and on-going professional development processes.

The ACPs had taken different training pathways into the role, including specific structured ACP Master’s level training and piecemeal alternative pathways pre-dating this qualification. There was widespread recognition that this made ACP recruitment and deployment challenging as their skill set, supervision and support needs could be so variable:

**7ACP**: *People who have done a prescribing module and pretty much nothing else will be sold as an ACP, right the way through to those of us who have got portfolios of practice, a full clinical masters and significant experience. We need to be very aware of that variety and sensitive to that during recruitment and selection processes. If you want essentially another GP, you either need to get a very good pre-existing, well established ANP [advanced nurse practitioner] and hit lucky or you are going to be sadly mistaken because if it is someone going through a development process, you have got to be able to commit to developing them.*

For PMs and GPs, the variability of ACP’s educational background created considerable confusion and complexities for ACP recruitment, and all stakeholders called for a standardized framework to support practice and training:


**21GP:**
*Because it is really confusing, I have to say, when you’re getting a doctor it is easy you just look at the performance list, as long as they are on the performance list you know they are a doctor and you just have to get your other bits of essential documentation, mandatory documentation but when it comes to a nurse it is really difficult to understand what they have done because sometimes the Master’s is relevant, sometimes the Master’s isn’t relevant it is so much more complex than a doctor.*


It was recognized that for many ACPs, their ability to perform well in the role was highly dependent upon the nature and level of support available at the individual practices:


**26PM:**
*She came directly from A&E and she needed teaching about general practice before she could become fully effective and we have done much of that training in house.*


For those who had undertaken a higher degree, there was a call for a more bespoke Primary Care Pathway as some Master’s degree programs were deemed to be too generic or too focused upon secondary care:


**4ACP:**
*It was very much driven towards hospital based ANPs [advanced nurse practitioners] and emergency care ANPs …so I decided not to do it because I wanted something that would be more specific to general practice, so I have tended to do standalone courses and e-learning.*


All practices reported a significant shortage of appropriate trained and skilled ACPs who would be able to ‘hit the ground running’, and those ACPs who had built up key expertise over many years were seen as a precious resource, but in short supply:


**11PM:**
*(ACPs) are moving around to different practices and then that vacancy is still there, it is just in a different practice. We find that you’re not getting new replenishments of staff with the skills that general practice needs.*


Once in post, difficulties in identifying and accessing continuing professional development (CPD) opportunities appropriate for advanced level practice were reported:


**16ACP:**
*It is difficult to get training that is aimed at ACPs because the training for the GPs is not appropriate for us, but we’re not practice nurses, [so] we need the next step up.*


The majority reported opportunities to be very ‘piecemeal’, requiring a considerable investment of the individual’s time in both sourcing and accessing any opportunities.

Experiences of supervision, mentorship and support were also quite variable across practices. Practices with a few established ACPs tended to have clearer mentorship systems in place, with regular debriefing, training and discussion around ethical scenarios. In other practices, restricted GP capacity was acknowledged to be a reason why limited support and supervision was available:


**13ACP:**
*I have been here five months and I think I have had ten minutes [of] supervision…which is utterly unacceptable.*


The majority of ACPs and PMs raised the need for a formal ACP support network to communicate and to share practice and learning with other ACPs:


**5PM:**
*They need to have a support network, whether that be a WhatsApp group or a meeting every quarter where they go together and they talk and discuss what is going on, so they can share education, knowledge and back each other up.*


Where ACPs were the only one in a practice, the need for a support network was seen to be essential:


**10ACP:**
*It would just be nice to integrate with some other ANPs I am the only ANP here…I haven’t got any other support other than the GP.*


It was noted that whilst such formal support was available in the region for GPs and for practice nurses, there was nothing specific for ACPs:


**7ACP:**
*We generally have a network support type structure within for practice nurses, we have the same in place for GPs, but we run the risk that the unique needs of ANPs and ACPs may fall between the crack, between the two of them.*


#### 3.3.4. Acceptance and Implementation of ACP Roles: Supporting a Culture Change

This theme explores the integration of ACPs into primary care, and looks at how the role was perceived to have been accepted and implemented, demonstrating a need for a cultural shift in the understanding of roles and practice organisation.

On the whole, all respondents were positive about the ACP role, and the contribution that ACPs brought to the practice was highly valued. In most practices, all interviewees were confident that ACPs knew and practiced safely within their level of clinical competence, referring onto GPs when necessary. There was very little resistance to the ACP role reported, with GPs and PMs noting in particular that any initial apprehension quickly disappeared once the benefits of the role became clear. The key to a smooth integration was developing a mutual understanding of the role:

**28GP**: *They are not doctors, they are nurses. They haven’t done the seven years training that we have done and I think that is largely true and in the way it works in our practice is that the ACPs recognise that they are very well qualified nurses and whilst they see many of the same patients that the doctors do, they recognise that the doctor needs to be there to support. And likewise, the GPs now recognise just how much work the ACPs can do and it is like a mutual respect situation and mutually supported situation.*

As described in the previous themes, the main challenges to role implementation were related to the variability of ACPs’ experience and skill set. This meant that their scope of practice needed to be individually negotiated within each practice and for each role:


**13GP:**
*If they feel they are not competent in an area then they don’t see those patients so it can very specific to that individual. So, it is tailored to the individual, I suppose then, isn’t it? We don’t say ‘right you have got to see everything’ and leave them to it as I suspect sometimes happens at some practices—so it is very much a role you know we sit here as in a supervisory role—there to refer up to if needed.*


For ACPs, some of whom were relatively new to primary care, this meant working to understand the limits to their own practice and to communicate this to their employer and team members, in order to ensure clear understanding and appropriate accountability and support for their workload, as well as to set up clear professional boundaries:

**25ACP**: *I am responsible for taking history, examining, coming up with a plan, obviously agreeing that with the patient. There are still patients where I just don’t know. It’s important to appreciate your limits, the extent of your knowledge and acknowledge that, I am not sure; but there are always GPs around that I can ask for advice.*

In making the decision to recruit ACPs, practices spoke of the need to embrace change and the need for a supportive culture to enable this. It was considered important for the whole team, including reception and administrative staff, to understand the ACPs’ role and scope of practice so that the whole system would support them. Several respondents described the need to train reception staff to ensure that patients/conditions were appropriately booked in according to their competence:

**4ACP**: *Initially there was quite a bit of anxiety over it, because they weren’t sure what I did/didn’t do.*

The participants generally felt that patients were positive towards the ACP role, and that they accepted the role and were satisfied with it. It was noted that in some cases, this had taken some time, and that there was some initial reluctance evident in many practices:


**6PM:**
*When we first took her on, we had to find a way of explaining to patients that she wasn’t a GP. We ended up where we actually gave a script to our reception staff that we prepared. Just a brief explanation so it was easier for them to actually explain to patients and try and answer their questions when they first rang up.*


The practices utilized multiple methods to educate patients about the ACP role, including an automated phone system to advise on the mix of clinicians, newsletters and website updates. It was seen that once patients better understood the role, knew that ACPs were able to prescribe medications and had gotten to know the individuals concerned, they seemed better able to make judgments about the role themselves and were therefore more positive:

**16ACP**: *We have many patients who prefer actually to see the ACPs rather than the GP, the feedback that I get is that ACPs are more thorough, they listen, they have a more holistic approach to their work.*

Several participants suggested a need for greater awareness raising at a national level around the ACP role in general practice in order to educate the public about the clinical scope of the ACP role, and thus promote acceptance and engagement at a time of great service demand:


**13ACP:**
*ANPs are doing 80% of the work that a GP does—but who is it who is shouting out for us? the media, in particular, is all negative about anybody other than GPs working in general practice. No acknowledgement that we all bring separate but complimentary things to the table.*


In terms of practice governance, having GP and ACP mentors was another important factor in ensuring ACPs’ integration. Again, this required an understanding of the role, individual competencies and development needs, and implementing communications to support the change. In the case of one practice (with a number of ACPs), this included creating a regular staff meeting specifically for the ACPs to support and communicate with each other and a mentorship group supported by the GP mentor with bespoke training.


**23PM:**
*They don’t actually fit with the nurses, and they don’t actually fit with the doctors but they are an integral part of the team, so they need some kind of communication, meeting themselves as well.*


Where such systems had not been developed, it could lead to ambiguity over the status of ACP (in terms of whether it was seen as a medical role or as a nursing role) and presented a dilemma as to where advanced practice fits in terms of practice governance.

#### 3.3.5. Impact of the ACP Role: Three Domains

The participants discussed the impact of the ACP role in 3 domains—in terms of the service delivery and enhancement, staff cohesion and in terms of individual staff workload and wellbeing. Overall, despite some challenges, the ACP role was considered as making a positive impact in all areas. ACPs were seen to enable stability and continuity of care by an appropriate clinician, with particular appreciation reported by the smaller practices. None of the interviewees reported any adverse events or patient complaints relating to ACPs’ practice. The majority of the practices reported positive impacts on service accessibility, with ACPs reported to be increasing appointment availability and making a significant contribution in dealing with urgent, acute and ‘on the day’ high demands. Where ACPs had taken a strong lead on home visiting and care home management, this was seen to have significantly altered GPs’ workload:


**16ACP:**
*The ACP role has had a definite positive benefit and certainly for GPs were just getting absolutely bogged down with home visiting and a lot of them were not appropriate for the GPs to go out.*


A few practices were able to offer new or enhanced services made possible through specific clinical skills of ACPs, including eye specialist work and a minor surgery clinic.

The perceived cost effectiveness of employing ACPs was a pertinent factor for the sustainability of practices. Significant cost benefits were emphasized, and the roles clearly enabled the focus to remain on ensuring that appointments were available, and patients seen.

**11PM**: *You think how much you pay for a GP and how much an advanced nurse practitioner costs, you know it is probably 50% cheaper and when you think they can probably do 75% of the same work.*

It was noted that many of the ACPs were viewed to have improved morale across the practice due to workload and skills being shared. Some practices highlighted ACPs as providing accessible support to team members, especially to receptionists and practice nurses when they had clinical queries. Practice managers were reassured by the knowledge and skills of ACPs involved in formal recruitment, training and development of other staff. For many participants, the ACPs were perceived to be significant role models, especially to the practice nurses, providing insight to progressive career pathways and helping to raise professional aspirations and opportunities.


**9GP:**
*I feel that the nursing team is stronger for having them on board…I think the more junior members of the nursing team can find that encouraging, you know they have got some role models and people to provide a bit of mentorship.*


In general, relatively high levels of job satisfaction and commitment were apparent from the ACPs. Professional and personal satisfaction was gained from using advanced level skills and knowledge in the treatment of patients, and the subsequent patient trust that developed:


**1ACP:**
*When I have taken patients who have undifferentiated diagnosis and I have diagnosed them and followed through with all the treatment, that is brilliant to really help people like that. You get to know your patients so well and they get to trust you and that actually is quite humbling is the trust that people put in you, it makes you want to do that job 150%.*


A number of ACPs had consciously moved from roles in emergency and urgent care, seeking and gaining a much better work–life balance in primary care. They reported feeling less pressurised by the volume of patients to be seen and felt a greater ability to switch off at the end of the day knowing that follow up and continuity of care remained possible. Some ACPs had moved from practices where they perceived work levels to be unsustainable and saw their role to be compromised.


**4ACP:**
*My other job was just getting stupid, it was just pressure, pressure, pressure, and I felt I was being used more for a substitute GP. Here there is that much support around me, I don’t feel stressed at all.*


In several cases, however, ACP stress levels were particularly high. High workload, high patient expectations, isolated working conditions and short 10-min appointment times were commonly cited factors that appeared to be associated with increased stress. This was more notable in the smaller practices where ACPs worked alone:


**13ACP:**
*Once you start to feel overwhelmed your cognitive process just aren’t working properly and that then becomes a real concern for me because to me patient safety is the most important thing. I do enjoy the job, I do find it very stressful. It has got a little bit better recently, I don’t know whether that is just because I am bedding down here and just getting used to the workload but certainly for 3 or 4 four months, I mean, I have had to stop wearing eye makeup because most days I have been in tears about something and I am quite a resilient person, but you know when I have got 57 patient to see and no end in sight, what is going on?*


The impact of appointment times was significant to ACP wellbeing and job satisfaction. Participants reported variable appointment lengths, with some having 10-min and some having 15-min appointment times. One practice offered more flexibility for clinicians to manage their own timings. Some reported 15–20 min initially in their training period, which then reduced to 10 min once they had gained more experience. The positive impact of the longer 15-min appointments was frequently reported.


**22ACP:**
*Our local guideline does say to us 15 min. I do feel that that gives me the opportunity to explore patients thoroughly and that does give me greater job satisfaction.*


GPs reported many benefits from sharing workloads with the ACPs. Some GPs described their increased managerial responsibilities (from supporting and mentoring ACPs) in a positive way, particularly those that had pioneered and led the workforce development and integration of the roles across practices. Other GPs acknowledged their need to commit more time to supervising and mentoring the ACPs. Some GPs perceived that ACPs had altered their case load so that they now had increasing numbers of complex and chronic cases. Whilst this was generally perceived to be more appropriate for the GP skill set, the fact that increasingly complex patients still needed to be dealt with in a 10-min consultation time led to a more intense and stressful work experience:


**17GP:**
*What I have found as we have employed more nurse practitioners…is that the complexity of the GP clinics goes up. When I was first starting, the clinic would be punctuated by quite a high number of sore throat, cough, sticky eye—you know, low challenge consultations that can be dealt with really, really quickly. And now a lot of the more simpler cases tend to see the nurse practitioners, so the relative complexity of a GP clinic is higher…But of course we have not said ‘oh GP’s now have 15 min appointments’ or anything like that. You know, we are still seeing people on 10 min appointments so that poses a real challenge.*


## 4. Discussion

### 4.1. Bespoke Roles and Bespoke Processes

The study confirms the existing calls for clarity, consistency and standardization of the ACP educational pathway and role [25]. The findings clearly show how the variability of ACPs’ capabilities leads to uncertainty for colleagues, and requires bespoke (and thus potentially inefficient) employment, deployment and support processes. This variability stems from the diverse educational and clinical backgrounds of the practitioners, varied length of experience as ACPs in primary care and local policy variations in terms of what ACPs were, and were not, sanctioned to do. Given the current imperatives for service re-design in primary care, this variability poses an obstacle to the implementation of system-wide innovations that could be widely applicable across the sector [39]. It is clear that a more standardized and sector-specific training pathway is needed. The recently released national framework for ACP [27] and the specific primary care standard [31] is greatly welcomed, and makes an important contribution to creating a more streamlined workforce and associated governance structures. As this framework is rolled out, it will be important to evaluate its impact.

### 4.2. Role Acceptance and Visibility

The data from this project suggests that, in the geographical region where the evaluation was carried out, there appeared to be a widespread acceptance of, and appreciation for, the ACP role. This is in contrast to many studies that highlight professional boundary tensions and role conflicts (both between GPs and ACPs as well as between practice nurses and ACPs) [38,39,57]. A few respondents noted some initial apprehension from GPs about the ACP role, but this was represented as a historical issue. Thus, despite the challenge to clearly delineate the role (posed by the variability in background and experience described above), the need for, and value of, ACPs was evident. Our findings suggest that the transition to ACP-delivered care can create initial uncertainties for practice staff (e.g., receptionists) and patients, requiring a degree of culture change. Over time and with support and education, however, these issues appeared to have been overcome and did not appear to pose significant ongoing obstacles. Relatively little work has been done in the UK on the patient experience of ACP roles in primary care, but the available evidence confirms our respondents’ suggestions that patient satisfaction with ACPs is generally high [58,59,60,61,62,63]. Nonetheless, the respondents in our study highlighted that the educational and awareness-raising work required to promote acceptance of the ACP role amongst the general public was primarily being implemented at the local practice level. The lack of a wider national advocacy for the role was seen to hinder public understanding and acceptance. In order to achieve a much wider public understanding of the role of ACP in primary care and thus create an enabling environment for practices to introduce this cadre, more needs to be done at the national level. For example, media campaigns and other forms of NHS messaging could depict general practice in terms of the range of staff that contribute there, rather than focusing primarily on the GP role.

### 4.3. Role Realization

An important finding of this study is that some ACPs are perhaps not yet working to their full potential. The study showed that ACPs are, understandably, primarily operating within the ‘clinical pillar ‘of their role. This is where their contribution was most recognized and where they themselves (as well as the GPs and PMs) highlighted their own development needs. It appeared that their capabilities and potential within the other 3 pillars, especially research and education, were less visible and less well recognized. This is consistent with the international literature in this area which has focused on documenting barriers to role transition and demonstrating the clinical effectiveness and safety of the role rather than explicitly considering the other pillars [38]. As a result, there is a lack of research on ACPs’ potential for leadership in primary care, or on their contributions in terms of education and research/quality improvement [64,65,66].

In the current study, the clinical emphasis appeared to have been partly due to the overwhelming pressures on services which meant that seeing patients was the top priority for ACPs and for practices overall. Thus, creating space within day-to-day service pressures for ACPs to develop other aspects of their role is a key challenge. Our findings suggest that there is a need to release ACP time to realize the full potential of this job role.

### 4.4. Staff Development, Workload and Wellbeing

Following on from the point above, for some ACPs, intense workload pressures were creating a stressful working environment, and some felt they had little support in the workplace. All participants groups recognized the need to have structures in places to provide mentorship, supervision, support and continuing professional development for ACPs, especially for those working in smaller practices. This is a finding that has not previously been extensively explored in existing literature, although one recent study conducted in another part of the UK reached the same conclusion [39].

A possible explanation for our findings is that whilst ACPs viewed their role as a step towards career enhancement, GPs and practice managers had a more pragmatic perspective, seeing the role as ‘filling the gaps’ in the GP workforce. There is a potential danger that such disjunctures in role conceptualization may, over time, lead to burnout and job dissatisfaction within the ACP workforce unless more attention is paid to their overall development needs. However, our study also highlights that it is unrealistic to place the onus for ACP support and career development provision on individual general practices, some of which are small. Our study suggests the need, therefore, for a system wide approach (e.g., through primary care networks) to ongoing ACP education and career development.

The study also showed that some GPs reported more stress due to a perception that their caseloads had become more complex as a result of ACPs taking on more of the routine aspects of patient care. Similar findings have been reported in other studies exploring GP views on the impact of ACP roles [60,67,68,69]. Prior work in this area has been mainly qualitative [37], and there is a need for further studies to examine the impact of changing skill mix on workloads in order to inform appropriate service design.

### 4.5. Recommendations for Future Research

Our evaluation suggests that more research is needed in the following areas:To explore the implementation and impact of the new ACP primary care nursing standard in the UK [31].To explore in more detail the workload and wellbeing impacts of ACP roles in primary care on different stakeholder groups, and to explore ways in which ACP wellbeing and work-related stress management can best be managed and supported. There is a particular need to explore more fully the influence of appointment times and patient numbers on the experience of ACPs at different stages in their career journey, and explore strategies that can be used to manage workload more effectively.To explore ways in which ACPs are enacting their role within the 3 non-clinical pillars to make their full contribution to primary care more visible and to more explicitly highlight the key development needs in these areas.To explore patient and carer views and experiences of different ACP roles in primary care.

Our study showed a great variability in the training and length of experience of ACPs in primary care. This suggests that future research on the ACP role in the UK needs to explicitly consider such variations in background and seniority in terms of how they affect role impact and satisfaction [70].

### 4.6. Recommendations for Service Development

Many of the challenges highlighted by this study could potentially be addressed by measures to enhance consistency and clarity in educational preparation and role specification. These issues will hopefully be tackled by the UK’s new ACP primary care nursing standard [31]. In addition, there is a need to develop structures to support the challenges identified around workload, stress, wellbeing and the need for continuing professional development. New ways of providing such support across the primary care system are required [39].

### 4.7. Strengths and Limitations

The practice-based sampling approach adopted by this evaluation was important to illuminate issues and challenges from the perspective of multiple stakeholders and multiple practice contexts, thus enhancing the transferability of the findings. Nonetheless, the relatively small number of practices in the sample could be considered a potential weakness and, as suggested above, additional research is required to extend our insights further, perhaps by adopting an in-depth case study approach in order to be able to more fully illuminate the organisational issues that influence ACP role enactment. Other potential weaknesses were the omission of the patient/carer perspective and the fact that the evaluation relied solely on qualitative data. Future work may benefit from additional methods and avenues of enquiry such as measures of workload, case complexity or cost effectiveness.

## 5. Conclusions

This study was an in-depth formative evaluation of ACP role implementation in diverse practice settings across one region of the UK. It showed a high degree of acceptance of the ACP role and emphasized the important contribution ACPs were making to meet patient care needs in the context of high workforce pressures and high service demand in primary care. Although limited to a UK setting, the study findings resonate with issues identified in the international literature [38]. The full realization of the ACP role potential is impeded by variability in ACP capabilities and limited visibility of the ACP contribution within non-clinical domains of practice. Moreover, there is a need for greater focus on workplace support and continuing professional development to support wellbeing in this occupational group, as well as to optimize the ACP workforce. ACP roles in primary care need to move beyond a perception of filling the gaps in existing provision to enabling wider service transformation.

## Figures and Tables

**Figure 1 ijerph-17-04500-f001:**
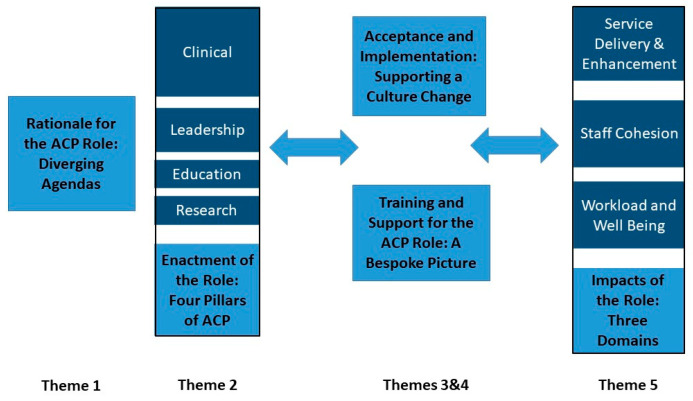
Themes.

**Table 1 ijerph-17-04500-t001:** General practitioner (GP) Practices.

County	Location	List Size (No. of Patients Registered)	No. of GPs	No. of ACPs
County A	City	11,000	7 part-time salaried GP’s. No partners	1
County A	City-Suburb	13,000	7 partners, 1 salaried and also GP registrars	1
County A	Semi-rural	6100	1 regular salaried GP and long term locums	1
County A	Small town	7100	1 GP lead, others are locums	2
County A	City	8200	2 GP partners; 2 salaried part-time GP’s	2
County B	Semi-rural	16,500	6 partners and 6 salaried GPs	2 (1 is a Partner). (also 2 trainees)
County B	Semi-rural	13,000	5 GP partners, 2 Salaried GP’s	1 (also a Partner)
County B	City	14,500	9 GP partners	2
County B	City	40,000	20 GPs	3

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
