# Peer review of "Advanced Clinical Practitioners in Primary Care in the UK: A Qualitative Study of Workforce Transformation"

_ijerph, 2020, doi:10.3390/ijerph17124500_

Round 1

Reviewer 1 Report

The manuscript concerns a very important issue: the role of Advanced Clinical Practitioners (ACP) in primary care. It is presented as a formative evaluation using a qualitative descriptive approach.

Major concerns:

  1. It is beneficial that ACPs, general practitioners, and practice manager are included in the study. However, if this is an evaluation it is important that these persons have enough experiences for serving as respondents. One line 197 it is stated that the experiences among the ACPs varied from 6 months to 16 years. There is only one respondent from each group from the nine practice sites included. And one GP declined to participate?
  2. The analysis of the data seems not to have taken the perspective within each practice site into account. Where there similarities among those coming from the same site?
  3. How has the three perspectives been analyzed? The patterns of excepts illustrates that the different categories have illuminated the researchers differently.: Among Practice Manager only 5 of 9 were cited (1 with 4 excepts and 1 with 2). Among the General Practitioners seven of eight were cited with each one one excerpt. Among the ACP all nine were cited with in total 26 excerpts (one with 5, one with 4, four with 3, two with 2 and one with 1 excerpts). What does this mean? That the external perspectives gave a less illuminative picture?
  4. Line 514: In the discussion it is pointed otu that the study both confirm and contradict existing literature. It is explicitly presented what confirms previous results but not what contradicts. But is it appropriate to discuss in this way using a qualitative study.

Minor concerns:

  1. Line 136: Does this mean that the study was "quick and dirty"?
  2. In the method description it is not stated who the interviewers were. Gender, age, occupation?
  3. Line 173: Those interviewed was not described with regard to background factors.
  4. Line 192 Is this group of GP Practices representative for the UK situation?
  5. Line 202 the number of patient seen per day varied from 22 to 57. This information could be used to compare different working conditions and its relationship with core aspects of the study.
  6. Line 205: The length of ACP appointments varied. Does this have a relationship with core aspects of the study.
  7. The results are interesting. Is it possible to develop a figure where the content of table 2 is included as well as the four pilars and three domains.
  8. Line312: ANP is not explained.
  9. Line 621: in-depth formative evaluation - is it enough to have included just nine GP Practices in the study?

The supplementary materials were excellent and informative.

Author Response

Thank you for your comments. Please see the attachment where our full response is provided.

Reviewer 2 Report

The manuscript submitted for review is an interesting and thoroughly prepared study. The authors show a good knowledge of the subject discussed and efficiency in achieving the adopted research goals. As a reviewer, I don't find any elements in the presented study that would require a more significant correction. Of course, the sample included in the survey was relatively small, but in this respect it does not deviate from the qualitative research standard. In addition, it is important to emphasize the high quality of the collected material analysis.

The authors of the study are aware of its limitations, hence, among others they avoid drawing too far-reaching or too generalized conclusions. At the same time, they recommend some interesting directions for further research that will allow to develop knowledge in the discussed problem area.

For my part, I can recommend the publication of a reviewed manuscript without any reservations.

Author Response

(The authors gave the same response as above.)

Reviewer 3 Report

I have read the paper: "Advanced Clinical Practitioners in Primary Care in the UK: A Qualitative Study of Workforce 4 Transformation", which is an excellent paper, well written, interesting, clear, rigorous.

I only have one commentary (only if the authors consider appropriated), I think that the results section, principally where you present the quotes of the interviewees, is too long and difficult to follow, maybe a table it would help to synthesize the findings and quotes.

The other minor thing is that in line 322 you use this numeration: "3.2.2 Rationale for the ACP Role: Divergent Agendas" and in line 348 you used it: "3.2.1 Enactment of the ACP Role: The Four Pillars". 

Excellent paper, I enjoyed reading it.

Author Response

(The authors gave the same response as above.)

Round 2

Reviewer 1 Report

The manuscript is now much improved. You responded to my comments in a qualified manner. The tension between being a formative evaluation and a qualitative study is discussed but not easy to resolve. Personally I have become more faviourable to mixed methods approaches especially when the study has an evaluative aim.

The information on researchers and participants illustrates the gender inbalance in certain research fields

Good luck with your further research endeavour.